# Pairing medical students on the wards: A multi-site analysis of pairing effect on clerkship performance

Krishan K. Sharma[1], Yuchiao Chang[2], Eli M. Miloslavsky[3]*

1 Division of Cardiology, Department of Medicine, Icahn School of Medicine at Mount Sinai in New York, New York, NY, United States of America, 2 Department of Medicine, Massachusetts General Hospital, Harvard Medical School in Boston, Boston, MA, United States of America, 3 Division of Rheumatology, Allergy and Immunology, Department of Medicine, Massachusetts General Hospital and Harvard Medical School in Boston, Boston, MA, United States of America

* emiloslavsky@mgh.harvard.edu

## Abstract

### Background

Medical students are often paired together on clinical teams during their clerkships. While this practice has multiple potential positive effects, evidence suggests that most students feel that their evaluation is impacted by the other student. This perception negatively impacts the learning environment. We set out to determine whether paired students had a measurable effect on each other's clerkship grade during the medicine sub-internship.

### Method

We examined 186 4th year student-pairs during the required medicine sub-internship at 3 hospital sites of Harvard Medical School from 2013–2017. Chi-square tests were used to determine whether pairing impacted the final clerkship grade. Subsequently we examined whether pairing impacted the sub-internship performance stratified by students' 3rd year core medicine clerkship grade to account for prior performance.

### Findings

We found no significant deviation between the expected and observed distribution of student grades (chi-square 1.9, p = 0.39) among 186 student pairs, suggesting that pairing had no meaningful effect on the sub-internship grade. We also saw no significant effect of pairing when controlling for prior internal medicine clerkship performance (chi-square 10.9, p = 0.53).

### Conclusions

Despite concerns that students on the same medical team may impact each other's performance evaluation, our exploratory study demonstrated no significant effect of student pairing on grades in a medicine sub-internship. Further study of the complex relationship between students on a medical team are warranted to optimize this common practice and enhance the learning environment.

**Data Availability Statement:** The datasets used and/or analyzed during the current study are available on Mendeley Data DOI: 10.17632/tn2587rxtt.1.

**Funding:** This study was supported by a grant from the Massachusetts General Hospital Center for Education Innovation and Scholarship which supported funding for gift cards and publication fees. The funders had no role in study design, data collection and analysis, decision to publish, or preparation of the manuscript.

**Competing interests:** The authors have declared that no competing interests exist.

## Introduction

Clinical clerkships represent some of the most educational and transformative experiences in undergraduate medical education [1]. Outside of the standardized elements of a taught curriculum, clerkship education is situated within the learning environment, taking place within the relationships between students, patients, and physicians, supported by informal elements of a learned curriculum [2]. At some medical schools, medical students are often paired together during their clerkships, where two or more students will work with the same medical team and be evaluated by the same group of resident and attending physicians. We recently reported that the majority of final year medical students perceive multiple positive impacts of pairing including effects on learning, adjustment to and enjoyment of the clerkship, wellness and the overall clerkship experience. However, the majority of students perceived that pairing impacted their clerkship evaluations and stress related to grading and evaluation was cited as a major negative impact of pairing [3].

Despite student concerns that pairing affects their grade, the effect of pairing on clerkship evaluations has not been previously investigated. Understanding the effect of pairing is important to enhance the clerkship learning environment. For example, prior studies of group-based learning modalities, such as team-based learning and case-based collaborative learning, utilized for pre-clinical curricula, have demonstrated that grouping medical students of different performance levels can often benefit low-performing students without hindering high-performing students [4, 5]. If this finding extends to the clerkship environment then pairing may become a tool to enhance student performance. Conversely, if student grades are negatively impacted, then pairing should be avoided in high-stakes clerkships. Finally, if no effect of pairing is seen, this may be the first step to alleviating student concerns about pairing impacting evaluation. We sought to begin exploring this relationship by analyzing whether paired students on a medical sub-internship team had a measurable effect on each other's clerkship grade.

## Methods

### Setting and participants

We performed a retrospective study examining students' grades for the required medicine sub-internship at Harvard Medical School (HMS) from 2013–2017 at 3 HMS sub-internship sites: Beth Israel Deaconess Medical Center (BIDMC), Brigham and Women's Hospital (BWH), and MGH. Our null hypothesis was that student pairing, regardless of their prior internal medicine clerkship performance, has no effect on their medicine sub-internship grades. Therefore, of the 232 student pairings during the study period we excluded 46 pairs containing visiting students from medical schools outside of HMS, or MD-PhD students who completed their internal medicine clerkship prior to entering their PhD. In addition, there were 46 instances of three medical students paired together on the same medical team where we included only the student with the highest sub-internship grade and student with the lowest grade among the triad for our primary analysis. The total number of students analyzed was 372. (Fig 1).

We chose to utilize the medicine sub-internship because it is the only required clerkship at Harvard Medical School HMS where students work in a single pair throughout the entire clerkship and are evaluated solely based on their clinical performance. We did not examine other clerkships because students either worked alone or were paired with multiple different partners during the rotation. In addition, other required clerkships utilized shelf exams in their grading, further hindering the evaluation of the pairing effect on performance.

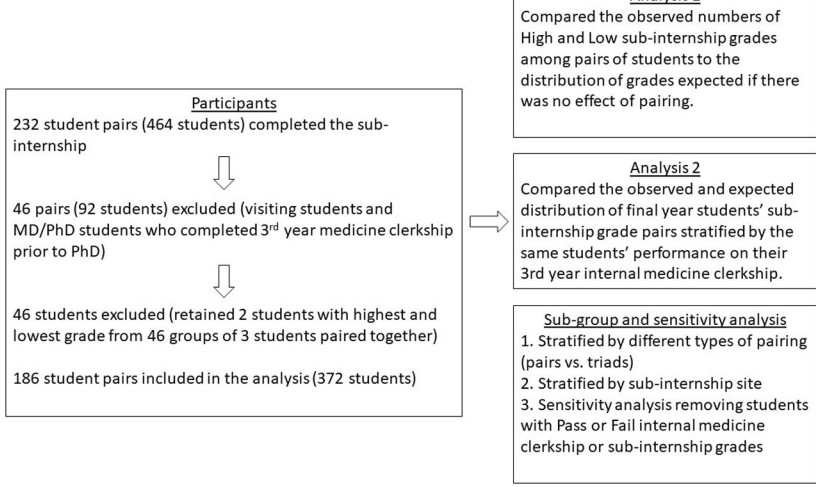

**Fig 1. Participant flow and analysis.**

The medicine sub-internship at HMS is a required four-week clinical experience where students in their final year of medical school are assigned to a general medical ward and function in a capacity similar to that of a first-year intern. During the study period of 2013–2017 students at all 3 sites were paired with one or two other students. This pairing generally replaced a single intern. All students were randomly assigned the same partner student(s) for the duration of the medicine sub-internship. "Pairing" was defined as students working with the same medical team of resident and attending physicians, in addition to participating in didactics together. Paired students worked on the general medicine service and would separately assume responsibility for their own patients, with the exception of one site (Massachusetts General Hospital [MGH]), where medical teams use a team-based care model, and responsibility for patient care is shared among all members of the team.

Sub-internship grades were assigned by the clerkship director at each site based on written evaluations by faculty and residents who worked directly with the student. Each evaluator submitted an assessment of the student on each of the following 10 items: history taking, physical examination, fund of knowledge, patient management, clinical evaluation and management skills, interpersonal skills, presentation skills, professionalism, cultural/social/systems awareness and initiative and desire to learn. The clerkship director then assigned one of the following grades based on their assessment of the written evaluations: Honors with Distinction (HD), Honors (H), Pass (P), or Fail (F). There was no written examination. The internal medicine clerkship grade, which is utilized in this study to control for prior performance (taken by students in their 3rd year), is also based on the same global rating comprised of the same 10 items. However, in order to achieve the highest grade, the student had to meet a minimum threshold on the National Board of Medical Examiners (NBME shelf) examination during the time period examined.

## Analysis of sub-internship grade

During the period of the study HMS awarded grades as discussed previously: HD, H, P and F. The grading schema was changed for the class taking the sub-internship in 2018, therefore students completing the sub-internship from 2013 to 2017 were studied. Of the 372 students in the analysis, 234 (63%) received HD, 122 (33%) received H, and 16 (4%) received P or F.

Because less than 5% of students receive a grade of either P or F, we dichotomized student grades into "High" and "Low" categories, with High defined as a grade of HD, and Low defined as a grade of either H, P, or F. To test our hypothesis, we compared the observed distribution of High-High, High-Low, and Low-Low student pairs to the expected distribution based on chance alone. (Fig 1).

## Analysis of sub-internship grade by prior performance in medicine core clerkship

To further address our research objective of determining how higher or lower-performing students affect each others' performance when paired together, we conducted an analysis of sub-internship grades that accounted for students' prior performance in their third year core internal medicine clerkship. We defined "high-performing" students as those who received a grade of HD on their internal medicine clerkship 152 (41%) and "low-performing" students 220 (59%) as those who received a grade of H, P, or F. Of note, the grade distribution for the medicine sub-internship is different than the medicine clerkship with more High grades awarded in the sub-internship. To test our hypothesis, we compared the observed distribution of High-High, High-Low, Low-High and Low-Low student pairs to the expected distribution based on chance alone.

### Statistics

Our main analyses were conducted using chi-square goodness-of-fit tests. The chi-square goodness of fit test is used specifically to analyze how the observed value differs from the expected value. We first compared the observed numbers of High and Low sub-internship grades among pairs of students to the distribution of grades expected if there was no effect of pairing. To investigate whether prior performance enhanced the pairing effect (e.g. a previously high performing student paired with a previously low performing student), we compared the observed and expected distribution of final year students' sub-internship grade pairs stratified by the same students' performance on their third year internal medicine clerkship. Additionally, we compared the probability of getting a High sub-internship grade between those paired with a student who had a High third-year internal medicine clerkship grade and those paired with a student who had a Low third-year internal medicine clerkship grade using Pearson chi-square tests.

We further conducted two subgroup analyses: (1) stratifying analyses by different types of pairing (pairs vs. triads) and (2) stratifying by site (team-based vs. non team-based care model). In addition, we conducted a sensitivity analysis removing students with Pass or Fail internal medicine clerkship or sub-internship grade from the analyses. We used SAS version 9.4 (Carey, NC) for all analyses, with statistical significance defined as a two-sided $P < 0.05$. The study protocol was approved by the institutional review board of HMS. The data analyzed had no personal identifiers and, given the retrospective nature, no consent was required.

## Results

We analyzed sub-internship grades from 372 students comprising 186 sub-intern pairs. We found that the distribution of students' sub-internship grade pairs was similar to the expected number based on random pairing ($p = 0.39$, Table 1), suggesting that students' grades during the sub-internship are not affected by their partners' grades when third year clerkship grades are not taken into account. Results were similar among the three sites.

To investigate whether students' prior performance affected student grades when paired, we utilized third-year core internal medicine clerkship grades as a proxy for prior performance. We first examined whether students who had high and low performing grades on their core third year clerkship were randomly paired together on sub-internship teams and found

**Table 1. Paired student grades for the medicine sub-internship.**

| Pairing | Observed | Expected* | P value |
|---|---|---|---|
| Both High | 78 | 73.6 | |
| One High, One Low | 78 | 86.8 | |
| Both Low | 30 | 25.6 | |
| Total | 186 | | 0.39** |

* Expected numbers were calculated based on Probability of High sub-internship grade of 0.63

** Chi-square goodness-of-fit test with 2 degrees of freedom; chi-square value = 1.9.

that the pairing was indeed random (*p* = .65). We then compared the observed and expected distribution of sub-internship grade pairs stratified by students' performance on their core third year internal medicine clerkship. (Table 2) Of note, the order of pairing in this analysis is important to determine its effect because a high performing student's effect on a low performing student and vice versa differ, therefore both High-Low and Low-High categories were analyzed. We found no significant deviation from our observed pairings to the expected values based on conditional probability (*p* = 0.53, Table 2).

To further corroborate this result, we found that among those with high internal medicine clerkship grade, 77% had a high sub-internship grade when they were paired with another high grade student compared to 83% when students were paired with a low grade student (*p* = 0.39, Table 3). Among those with low internal medicine clerkship grade, 45% had a high sub-internship grade when they were paired with a high grade student compared to 55% when students were paired with another low grade student (*p* = 0.17), further suggesting no meaningful effect of pairing.

In the sensitivity analysis stratified by different types of pairing, we found similar results between students paired in two and students paired in three. When stratified by site, there were no significant differences found. We observed similar results when the students with Pass or Fail grades were removed from analysis.

## Discussion

We examined the effect of student pairing on grading, finding that there is no significant effect of a given student's clinical performance on their partner's performance during the medicine

**Table 2. Paired student grades for the medicine sub-internship, stratified by third year internal medicine clerkship performance.**

| Observed no. (Expected no.)* | | Sub-internship grade pairs | | | | | P value |
|---|---|---|---|---|---|---|---|
| | | High-High | High-Low | Low-High | Low-Low | Total | |
| **Internal medicine clerkship grade pairs** | High-High | 17 (18.3) | 5 (4.3) | 4 (4.3) | 2 (1.0) | 28 | 0.76** |
| | High-Low | 19 (18.4) | 19 (18.0) | 1 (4.3) | 6 (4.3) | 45 | 0.34** |
| | Low-High | 20 (20.8) | 3 (4.9) | 22 (20.4) | 6.0 (4.8) | 51 | 0.76** |
| | Low-Low | 22 (15.8) | 9 (15.5) | 15 (15.5) | 16 (15.2) | 62 | 0.16** |
| | Total | 78 | 36 | 42 | 30 | 186 | 0.53§ |

* Expected numbers were calculated based on the probability (High sub-internship grade | High internal medicine grade) = 0.81 and probability (High sub-internship grade | Low internal medicine grade) = 0.50. For example, the probability of getting a High-High sub-internship grade pair from a High-Low internal medicine grade pair is 0.81 x 0.51 = 0.41 and expected number of pairs is 45 x 0.41 = 18.4.

** Chi-square test with 3 degrees of freedom; chi-square value = 1.2, 3.4, 1.2, 5.2 for the 4 rows, respectively.

§ Chi-square test with 12 degrees of freedom; chi-square value = 10.9.

sub-internship. Despite student pairing being common on clinical clerkships, this is the first study to our knowledge assessing the impact of this practice on student grades. In addition to our earlier work demonstrating multiple perceived impacts of pairing, prior research has demonstrated that peer relationships support the moral development of medical students, peer-led teaching improves knowledge acquisition in both clinical and pre-clinical settings, and student peer observation improves clinical skills [6–8]. Therefore, in order to enhance the clerkship learning environment it is important to maximize the positive impacts of pairing while minimizing perceived negative effects.

Our prior work has demonstrated that stress related to grading and evaluation was cited as a major negative aspect of pairing students in a clerkship. However, the results of this study indicate that there may be a discrepancy between the widespread student perception that pairing impacts their evaluation and real-world effect of pairing on grading [3]. The educational environment of clerkships encompasses the physical, social, and psychological contexts in which students learn, including interactions with faculty and peers, along with informal and hidden curricula [9]. School-related stressors, including academic pressure and grading, have been shown to negatively impact the learning environment and students' well being [10]. Educators must continue to work towards optimizing the educational setting of clerkships, as the learning environment influences students' professional development and identity [2]. Unsupportive learning environments, including non-collaborative and competitive settings, have been independently associated with student burnout and distress, and can potentially alter critical components of student motivation, including their sense of safety, belonging, and self-esteem [11, 12]. Increased concern over evaluations and grading have also been associated with student burnout, and may potentially impair student learning by increasing extraneous cognitive load [13, 14]. Therefore, a better understanding by both students and faculty of the real and perceived effects of pairing has the potential to enhance the clerkship learning environment. The findings of our study may help students and faculty limit the negative concerns regarding pairing, stemming from stress regarding comparative grading, and enhance the positive aspects of pairing such as effects on learning, adjustment to the clerkship, enjoyment, wellness and the overall clerkship experience.

Our findings differ from prior investigation of classroom-based group learning during medical school, where grouping students of different performance levels has demonstrated a benefit to low-performing students without hindering high-performing students [4, 5]. This is not surprising as the clinical environment differs significantly from classroom-based courses. For example, there may be more limited peer interaction and peer-assisted learning within the clinical environment, when compared to formalized, classroom-based group learning, amongst other factors. Further investigation of peer-assisted learning on clinical clerkships is warranted as well as efforts to limit competition amongst students as it relates to patient exposure and evaluations [7].

**Table 3. Sub-internship high grades by internal medicine clerkship performance.**

| Internal Medicine Clerkship Grade of Student 1 | Internal Medicine Clerkship Grade of Student 2 | N | Sub-internship High grade |
|---|---|---|---|
| High | High | 56 | 43 (77%) |
| | Low | 96 | 80 (83%) |
| Low | High | 96 | 43 (45%) |
| | Low | 124 | 68 (55%) |
| Total | | 372 | 234 |

Our results also inform the issue of evaluator cognitive bias. Substantial variability exists among clinical evaluators with respect to the reliability, accuracy, and validity of assessments made when directly observing trainees [15–17]. Moreover, the halo effect and multiple other biases have been described affecting how a given individual may be evaluated [18, 19]. If two students in a given pair are of different performance levels, students may be reluctant to be evaluated by the same medical team due to fear of comparison [20]. While our analysis puts into question the role of the halo effect and other comparator biases on evaluations of paired students, we cannot rule out the possibility that a comparator bias exists and is balanced by other factors, such as peer-teaching and peer support within a student pair. Further investigation of this dynamic as it relates to student pairs through qualitative methods may be of interest.

We acknowledge several limitations to our study. First, this study represents the findings of a single medical school and was retrospective in nature, with no true control group of students who were unpaired. In addition, the findings of this study are dependent on the interplay between two students on a medical sub-internship, and thus may not be fully generalizable to other clerkships or medical schools, where the interactions of paired students may differ. However, it was not possible to isolate the impact of pairing in other clerkships due to multiple or inconsistent pairing and effect of factors other than clinical evaluation on grades. Sixty three percent of the students received the top grade on the sub-internship, thus potentially limiting the sensitivity of our study. However, in the majority of U.S. medical schools the sub-internship grades skew towards a high percentage of students getting the top grade as compared to core clerkships and the HMS sub-internship has a relatively lower percentage of students receiving the top grade than other U.S. medical schools making it optimal for investigation. Furthermore, we cannot exclude the possibility that a small effect size of pairing on grades exists. However, our study included over 370 students at 3 different sites and we feel the study was of sufficient size to exclude a meaningful effect. Additional data outside of our study period could not be included due to data availability and curriculum changes. Furthermore, we did not have data regarding the sub-items that comprise the final grade and therefore could not evaluate any potential impact of pairing on specific sub-items. The strengths of our study include multiple years of grading data from three different hospitals in a clerkship with consistent student pairing throughout its duration.

Future work in this area could include gaining a better understanding of student perceptions and interactions within a pair through qualitative methods, examining the effects of pairing on performance and the learning environment in other clerkships, and evaluating the effect of interventions such as enhanced opportunities for peer-teaching and collaboration on student pairs.

## Conclusions

Our exploratory study demonstrated no significant effect of student pairing on grades in a medicine sub-internship, identifying a potential discrepancy between student perception and actual effect. We believe this work serves as an important step towards understanding the impact of pairing students together on clerkships. Further, it has the potential to inform clerkship design, student and faculty education regarding this phenomenon and interventions that enhance the effect of pairing, thereby enhancing the clerkship experience.

## Acknowledgments

The authors would like to acknowledge Alberto Puig, Meredith Atkins, Erik Alexander, Alex Carbo, Benjamin Davis, Mary Montgomery, Subani Chandra, Holly Gooding, David Hirsh, and Grace Huang for their expert input.

## Author Contributions

**Conceptualization:** Krishan K. Sharma, Eli M. Miloslavsky.

**Data curation:** Krishan K. Sharma, Yuchiao Chang, Eli M. Miloslavsky.

**Formal analysis:** Krishan K. Sharma, Yuchiao Chang, Eli M. Miloslavsky.

**Funding acquisition:** Eli M. Miloslavsky.

**Investigation:** Krishan K. Sharma, Eli M. Miloslavsky.

**Methodology:** Krishan K. Sharma, Yuchiao Chang, Eli M. Miloslavsky.

**Project administration:** Krishan K. Sharma, Eli M. Miloslavsky.

**Supervision:** Eli M. Miloslavsky.

**Writing – original draft:** Krishan K. Sharma, Yuchiao Chang, Eli M. Miloslavsky.

**Writing – review & editing:** Krishan K. Sharma, Yuchiao Chang, Eli M. Miloslavsky.

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
