## [Decision Letter · Decision Letter 0]

18 May 2022

PONE-D-21-36455

Pairing medical students on the wards: A multi-site analysis of pairing effect on clerkship performance

PLOS ONE

Dear Dr. Miloslavsky,

Thank you for submitting your manuscript to PLOS ONE. After careful consideration, we feel that it has merit but does not fully meet PLOS ONE’s publication criteria as it currently stands. Therefore, we invite you to submit a revised version of the manuscript that addresses the points raised during the review process.

We look forward to receiving your revised manuscript.

Kind regards,

Rano Mal Piryani, MBBS, MCPS, DTCD, MD, Fellowship in Med Education

Academic Editor

PLOS ONE

**Journal requirements:**

3. Please expand the acronym “EMM” (as indicated in your financial disclosure) so that it states the name of your funders in full.

4. "Thank you for stating the following financial disclosure:

“EMM - This study was supported by a grant from the Massachusetts General Hospital Center for Education Innovation and Scholarship which supported funding for gift cards and publication fees.”

5. PLOS requires an ORCID iD for the corresponding author in Editorial Manager on papers submitted after December 6th, 2016. Please ensure that you have an ORCID iD and that it is validated in Editorial Manager. To do this, go to ‘Update my Information’ (in the upper left-hand corner of the main menu), and click on the Fetch/Validate link next to the ORCID field. This will take you to the ORCID site and allow you to create a new iD or authenticate a pre-existing iD in Editorial Manager. Please see the following video for instructions on linking an ORCID iD to your Editorial Manager account: https://www.youtube.com/watch?v=_xcclfuvtxQ.

**Additional Editor Comments:**

Dear Authors

We have received the reports from our advisors on your manuscript

Based on the advice received, the Academic Editor feels that your manuscript could be reconsidered for publication should you be prepared to incorporate major revisions.

When preparing your revised manuscript, you are asked to carefully consider the reviewer comments which are attached, and submit a list of responses to the comments.

Your list of responses should be uploaded as a file in addition to your revised manuscript.

Best regards,

Reviewers' comments:

Reviewer's Responses to Questions

**Comments to the Author**

1. Is the manuscript technically sound, and do the data support the conclusions?

Reviewer #1: Yes

Reviewer #2: Yes

2. Has the statistical analysis been performed appropriately and rigorously? 

Reviewer #1: No

Reviewer #2: Yes

3. Have the authors made all data underlying the findings in their manuscript fully available?

Reviewer #1: No

Reviewer #2: No

4. Is the manuscript presented in an intelligible fashion and written in standard English?

Reviewer #1: Yes

Reviewer #2: Yes

5. Review Comments to the Author

Reviewer #1: Abstract (Methods) …….to assess whether students paired together on a medical team during their Internal Medicine sub-internship affected each other’s grade.

Comment: This is again repetition of objective) What statistical tests were applied (Missing). How they were examined (might be OSCE / MCQs) is not clear.

Introduction line 45

Clinical clerkships represent some of the most educational and transformative experiences in undergraduate medical education.1 Comment: No reference needed.

Statistics Line 134 -135

To investigate whether the effect of pairing students was dependent on prior performance, we compared the observed and expected distribution Comment: It was retrospective study, they had factual (true) data of 3rd and 4th year. They didn’t need; expected values of grades. Authors should go for other statistical tests (parametric if data normally distributed)

Setting Line 85-87

….which is comprised of the following 10 items:

Comments: There were 10 variables authors could observe the effect of pairing on these variables by comparing 3rd and final year students. If they do, it would be much better.

Analysis Line 96-97

Comment: Authors had data from 3 hospital sites of Harvard Medical school. They could compare the three sites using ANOVA.

Comment: ** Chi-square test with 3 degrees of freedom authors never write chi-square value? this is very important to mention its value. They should add two columns more in each table one for residual value and other for chi-square value

Comment: It is not clear whether students from 3rd year and 4th year were the same or different. Authors should clarify. Reading further it looks the students (3rd and 4th yr.) were the same. In this case Paired t test could be used to observe the difference in their grades.

Conclusions - Line 261

Our exploratory study demonstrated no significant effect of student pairing on grades in a medicine sub-internship. Comments ( it was retrospective study)

Reviewer #2: It was a pleasure to review this manuscript. The following comments are attached to improve this study.

The title and abstract are satisfactory.

The Introduction conveys key messages about context. However, the rationale of the study is unclear. Please elaborate more.

The methods have sufficient detail and flow logically but I have minor suggestions to improve clarity. Please provide details on how the students were paired other than randomization such as the low performer with the high performer and who assessed the students in three different sites. The authors need to explain the training and delivery method in three different sites. Also, the assessors’ qualifications, experience, etc.

The Discussion explores the Results satisfactorily. Please provide limitations and more appropriate implications of the study results.

The Conclusion reflects the Results well. Clarify the potential transferability of findings, expanding on what you already conclude about "this work serves as an important step towards evaluating the impact..."

You have written this paper well enough for me to understand what you were trying to do and why, what you did, what you found, what that means, and so what? My suggestions are relatively minor and in the spirit of crafting some parts better and clarifying potential confusion.

6. PLOS authors have the option to publish the peer review history of their article (what does this mean?). If published, this will include your full peer review and any attached files.

Reviewer #1: No

Reviewer #2: No

---

## [Author Response · Author response to Decision Letter 0]

29 Jun 2022

Editor reviewers and editorial team,

Thank you for your thoughtful review of our manuscript. We have incorporated the reviewer suggestions, as detailed below, which have significantly improved our manuscript.

Reviewer #1: 

Comment: Abstract (Methods) …….to assess whether students paired together on a medical team during their Internal Medicine sub-internship affected each other’s grade. This is again repetition of objective) What statistical tests were applied (Missing). How they were examined (might be OSCE / MCQs) is not clear.

Response: We thank the reviewer for this comment. We have eliminated the redundant statement and added the following “Chi-square tests were used to determine whether pairing impacted the final clerkship grade, accounting for prior performance.”

Comment: Introduction line 45 - Clinical clerkships represent some of the most educational and transformative experiences in undergraduate medical education.1 Comment: No reference needed.

Response: Thank you, we took out this reference.

Comment: Statistics Line 134 -135 - To investigate whether the effect of pairing students was dependent on prior performance, we compared the observed and expected distribution Comment: It was retrospective study, they had factual (true) data of 3rd and 4th year. They didn’t need; expected values of grades. Authors should go for other statistical tests (parametric if data normally distributed)

Response: Thank you for asking for this important clarification. The analysis we used appropriately tests our null hypothesis, that is that the actual grades that were observed did not differ from the distribution that would have been expected if pairing students had no impact on grades. We revised this section to state “We first compared the observed numbers of High and Low sub-internship grades among pairs of students to the distribution of grades expected if there was no effect of pairing. To investigate whether prior performance enhanced the pairing effect (e.g. a previously high performing student paired with a previously low performing student), we compared the observed and expected distribution of 4th year sub-internship grade pairs stratified by the same students’ performance on their 3rd year internal medicine clerkship.

With regard to the question of whether parametric tests should have been utilized, to determine whether the effect of pairing students was dependent on prior performance, we compared the observed distribution with the expected distribution from the independence assumption using a chi-square test. Table 1 shows the distribution of pairing based on students’ prior performance. Theoretically, the number of pairs in both high, one high one low, and both low categories are from a multinomial distribution. For that reason, we did not use a parametric test since data could not be considered as normally distributed. The results showed no significant difference between the observed distribution and the expected distribution if pairing was random.

Comment: Setting Line 85-87 -….which is comprised of the following 10 items: There were 10 variables authors could observe the effect of pairing on these variables by comparing 3rd and final year students. If they do, it would be much better.

Response: We agree with the reviewer that it would have provided more insight if the analysis could be done for all 10 components of the clinical performance. Unfortunately, the only data available to us was the global rating. Prior studies suggest that the global rating is a valid and important outcome measure, therefore we felt that for this exploratory study it would be an appropriate outcome. We included this in the limitations “Furthermore, we did not have data regarding the sub-items that comprise the final grade and therefore could not evaluate any potential impact of pairing on sub-items.”

Comment: Analysis Line 96-97 - Authors had data from 3 hospital sites of Harvard Medical school. They could compare the three sites using ANOVA.

Response: We thank the reviewer for this comment. We examined the observed and expected distribution by site and saw no difference. We have now added a sentence to Line 153.

Comment: Chi-square test with 3 degrees of freedom authors never write chi-square value? this is very important to mention its value. They should add two columns more in each table one for residual value and other for chi-square value

Response: We thank the reviewer for this comment. We debated between readability and thoroughness in presenting data in the two tables. The additional columns would have crowded the tables which are already quite wide. As an alternative, we have now added the chi-square values to the footnotes in each table (Line 157 and Line 178, and Line 179). 

Comment: It is not clear whether students from 3rd year and 4th year were the same or different. Authors should clarify. Reading further it looks the students (3rd and 4th yr.) were the same. In this case Paired t test could be used to observe the difference in their grades.

Response: Thank you for asking for this important clarification. This study included one cohort of students for whom we had both 3rd year medicine clerkship and 4th year medicine sub-internship grades. We do not compare the 3rd year to 4th year grades. Rather, we ask whether students that are paired together during their 4th year sub-internship impact each others’ 4th year grade. We use their 3rd year grade to ask whether prior performance can bring out the pairing effect in the 4th year (e.g. what if a pairing effect on grades is only seen when high and low performing students are paired together). We attempted to clarify this point further throughout the text.

Comment: Conclusions - Line 261 – “Our exploratory study demonstrated no significant effect of student pairing on grades in a medicine sub-internship.” - it was retrospective study 

Response: We highlighted the fact that it was a retrospective study as follows “Our exploratory retrospective study demonstrated no significant effect of student pairing on grades in a medicine sub-internship.” However we do note that if the study was done prospectively, the data would be unlikely to change given that grading and pairing should not be affected by the study.

 

Reviewer #2: 

Comment: It was a pleasure to review this manuscript. The following comments are attached to improve this study.

The title and abstract are satisfactory.

The Introduction conveys key messages about context. However, the rationale of the study is unclear. Please elaborate more.

Response: We thank the reviewer for this suggestion and have revised the second paragraph to elaborate on the rationale as follows “Despite student concerns that pairing affects their grade, the effect of pairing on clerkship evaluations has not been previously investigated. Understanding the effect of pairing is important to enhance the clerkship learning environment. For example, prior studies of group-based learning modalities, such as team-based learning and case-based collaborative learning, utilized for pre-clinical curricula, have demonstrated that grouping medical students of different performance levels can often benefit low-performing students without hindering high-performing students.4,5 If this finding extends to the clerkship environment then pairing may become a tool to enhance student performance. Conversely, if student grades are negatively impacted, then pairing should be avoided in high-stakes clerkships. Finally, if no effect of pairing is seen, this may be the first step to alleviating student concerns about pairing impacting evaluation. We sought to begin exploring this relationship by analyzing whether paired students on a medical sub-internship team had a measurable effect on each other’s clerkship grade.

Comment: The methods have sufficient detail and flow logically but I have minor suggestions to improve clarity. Please provide details on how the students were paired other than randomization such as the low performer with the high performer and who assessed the students in three different sites. The authors need to explain the training and delivery method in three different sites. Also, the assessors’ qualifications, experience, etc.

Response: Thank you for these comments. All pairing was random. High performing students were not intentionally paired with low performing students. We specify this by revising the sentence to read “All students were randomly assigned the same partner student(s) for the duration of the medicine sub-internship, making the effect of student pairing suitable for investigation.”

Medicine sub-internship responsibilities involve primary ownership of patient care as a first-year intern would be expected to do. Evaluation is based solely on clinical performance on the wards. The grade is assigned by the clerkship director based on evaluations from faculty and residents working directly with the student. We have added additional detail to the manuscript as follows: “The medicine sub-internship at HMS is a required four-week clinical experience where students in their final year of medical school are assigned to a general medical ward and function in a capacity similar to that of a first-year intern. ….. Paired students worked on the general medicine service and would separately assume responsibility for their own patients, with the exception of one site (Massachusetts General Hospital [MGH]), where medical teams use a team-based care model, and responsibility for patient care is shared among all members of the team. Sub-internship grades were assigned by the clerkship director at each site based on evaluation by faculty and residents who worked directly with the student.” 

Comment: The Discussion explores the Results satisfactorily. Please provide limitations and more appropriate implications of the study results.

Response: Thank you for this suggestion. We have enhanced the limitations section which now reads: “We acknowledge several limitations to our study. First, this study represents the findings of a single medical school and was retrospective in nature, with no true control group of students who were unpaired. In addition, the findings of this study are dependent on the interplay between two students on a medical sub-internship, and thus may not be fully generalizable to other clerkships or medical schools, where the interactions of paired students may differ. However, it is not possible to isolate the impact of pairing in other clerkships due to multiple or inconsistent pairing and effect of factors other than clinical evaluation on grades. Sixty three percent of the students received the top grade on the sub-internship, thus potentially limiting the sensitivity of our study. However, in the majority of U.S. medical schools the sub-internship grades skew towards a high percentage of students getting the top grade as compared to core clerkships and the HMS sub-internship has a relatively lower percentage of students receiving the top grade than other U.S. medical schools making it optimal for investigation. Furthermore, we cannot exclude the possibility that a small effect size of pairing on grades exists. However, our study included over 370 students at 3 different sites and we feel the study was of sufficient size to exclude a meaningful effect. Additional data outside of our study period could not be included due to data availability and curriculum changes. Furthermore, we did not have data regarding the sub-items that comprise the final grade and therefore could not evaluate any potential impact of pairing on sub-items. The strengths of our study include multiple years of grading data from three different hospitals in a clerkship with consistent student pairing throughout its duration.” 

We have also added several sentences to clarify the implications of our results as follows “The findings of our study may help students and faculty limit the negative concerns stemming from stress regarding comparative grading and enhance the positive aspects of pairing such as effects on learning, adjustment to the clerkship, enjoyment, wellness and the overall clerkship experience.3”

Comment: The Conclusion reflects the Results well. Clarify the potential transferability of findings, expanding on what you already conclude about "this work serves as an important step towards evaluating the impact..."

Response: We agree and elaborated on our conclusion as follows “Our exploratory study demonstrated no significant effect of student pairing on grades in a medicine sub-internship, identifying a potential discrepancy between student perception and actual effect. We believe this work serves as an important step towards understanding the impact of pairing students together on clerkships. Further, it has the potential to inform clerkship design, student and faculty education regarding this phenomenon and interventions that enhance the effect of pairing, thereby enhancing the clerkship experience.” In addition, we include a new paragraph in the discussion suggesting additional investigations in this area: “Future work in this area could include gaining a better understanding of student perceptions and interactions within a pair through qualitative methods, examining the effects of pairing on performance and the learning environment in other clerkships, and evaluating the effect of interventions such as enhanced opportunities for peer-teaching and collaboration on student pairs.”

Comment: You have written this paper well enough for me to understand what you were trying to do and why, what you did, what you found, what that means, and so what? My suggestions are relatively minor and in the spirit of crafting some parts better and clarifying potential confusion.

Response: Thank you for the positive feedback and suggestions which have improved the manuscript.

---

## [Decision Letter · Decision Letter 1]

3 Oct 2022

PONE-D-21-36455R1Pairing medical students on the wards: A multi-site analysis of pairing effect on clerkship performancePLOS ONE

Dear Dr. Miloslavsky,

Thank you for submitting your manuscript to PLOS ONE. After careful consideration, we feel that it has merit but does not fully meet PLOS ONE’s publication criteria as it currently stands. Therefore, we invite you to submit a revised version of the manuscript that addresses the points raised during the review process.

We look forward to receiving your revised manuscript.

Kind regards,

Rano Mal Piryani, MBBS, MCPS, DTCD, MD, Fellowship in Med Education

Academic Editor

PLOS ONE

Journal Requirements:

Reviewers' comments:

Reviewer's Responses to Questions

**Comments to the Author**

1. If the authors have adequately addressed your comments raised in a previous round of review and you feel that this manuscript is now acceptable for publication, you may indicate that here to bypass the “Comments to the Author” section, enter your conflict of interest statement in the “Confidential to Editor” section, and submit your "Accept" recommendation.

Reviewer #1: (No Response)

Reviewer #3: All comments have been addressed

Reviewer #4: All comments have been addressed

2. Is the manuscript technically sound, and do the data support the conclusions?

Reviewer #1: No

Reviewer #3: Yes

Reviewer #4: Yes

3. Has the statistical analysis been performed appropriately and rigorously? 

Reviewer #1: No

Reviewer #3: I Don't Know

Reviewer #4: Yes

4. Have the authors made all data underlying the findings in their manuscript fully available?

Reviewer #1: No

Reviewer #3: Yes

Reviewer #4: Yes

5. Is the manuscript presented in an intelligible fashion and written in standard English?

Reviewer #1: Yes

Reviewer #3: Yes

Reviewer #4: Yes

6. Review Comments to the Author

Reviewer #1: Abstract

Methods

Chi-square tests were used to determine whether pairing impacted the 29 final clerkship grades, accounting for prior performance.

Findings

Comments: let’s suppose, the Chi-square test is applicable (in fact, it is NOT), authors should mention its value in the abstract. They have mentioned in the results section. The alpha value of 2 df is 5.99 and their calculated value is 1.9 (Table 1 ), which means they should accept the null hypothesis. Right it is ok

The Results are inappropriate. Chi-square is applied when data is independent and nominal (Categorical).

Reviewer #3: You have written this paper well enough for me to understand what you were trying to do and why? My suggestions are relatively minor.

Abstract: add keywords at the end

Methods: this section will be clarified more if it has a figure or flow chart that show the process of the research , how you included the students and how you compare grades with previous years, and whether third year students are paired or individually.

Reviewer #4: Dear Authors,

Please check the following details:

Abstract:

1- Please explain more about the procedure of your work.

2- Please write the keywords based on MeSH.

Introduction:

This section was written very well.

Methods:

3- The second sentence was not written clearly; please modify it.

4- It would be better to write the number of paired students in the setting section.

5- Please explain more about the sub-internship grade in the setting section.

6- It would be better to write the participants section at the beginning of the methods section.

Results:

7- The description of table 2 was not precise and clear.

Discussion and Conclusion:

These sections were written well.

7. PLOS authors have the option to publish the peer review history of their article (what does this mean?). If published, this will include your full peer review and any attached files.

Reviewer #1: No

Reviewer #3: **Yes: **Nazdar Ezzaddin Alkhateeb

Reviewer #4: No

---

## [Author Response · Author response to Decision Letter 1]

7 Oct 2022

Reviewer #1: 

Reviewer comment: Abstract/Methods

Chi-square tests were used to determine whether pairing impacted the 29 final clerkship grades, accounting for prior performance.

Let’s suppose, the Chi-square test is applicable (in fact, it is NOT), authors should mention its value in the abstract. They have mentioned in the results section. The alpha value of 2 df is 5.99 and their calculated value is 1.9 (Table 1 ), which means they should accept the null hypothesis. Right it is ok

The Results are inappropriate. Chi-square is applied when data is independent and nominal (Categorical).

Author response: We thank the reviewer for this suggestion and have included the chi square values in the abstract. We felt the chi-square goodness of fit test is appropriate given that the conditions of categorical variables and random sampling were met.

Reviewer #3: 

Reviewer comment: You have written this paper well enough for me to understand what you were trying to do and why? My suggestions are relatively minor.

Abstract: add keywords at the end

Author response: We have added keywords to the manuscript submission based on MeSH terms. ([Education, Medical, Undergraduate], Clinical Clerkship, learning environment, student evaluation)

Reviewer comment: Methods: this section will be clarified more if it has a figure or flow chart that show the process of the research , how you included the students and how you compare grades with previous years, and whether third year students are paired or individually.

Thank you for this suggestion. We have included a figure detailing our methods (Figure 1).

Reviewer #4: 

Reviewer comment: Dear Authors,

Please check the following details:

Abstract:

1- Please explain more about the procedure of your work.

Author response: Thank you for this suggestion, we expanded the Methods section of the abstract as follows. “We examined 186 4th year student-pairs during the required medicine sub-internship at 3 hospital sites of Harvard Medical School from 2013-2017. Chi-square tests were used to determine whether pairing impacted the final clerkship grade. Subsequently we examined whether pairing impacted the sub-internship performance stratified by students’ 3rd year core medicine clerkship grade to account for prior performance.”

Reviewer comment: 2- Please write the keywords based on MeSH.

Author response: We have added the keywords as suggested

Reviewer comment: Introduction:

This section was written very well.

Methods:

3- The second sentence was not written clearly; please modify it.

Author response: We agree and have reworded this sentence as follows, “We did not examine other clerkships because students either worked alone or were paired with multiple different partners during the rotation. In addition, other required clerkships utilized shelf exams in their grading, further hindering the evaluation of the pairing effect on performance.”

Reviewer comment: 4- It would be better to write the number of paired students in the setting section.

Author response: Thank you for this suggestion. We merged the setting and participants section with the participants section first to have the number of students examined early in the methods section.

Reviewer comment: 5- Please explain more about the sub-internship grade in the setting section.

Author response: We elaborated on the sub-internship grade as follows. “Sub-internship grades were assigned by the clerkship director at each site based on written evaluations by faculty and residents who worked directly with the student. Each evaluator submitted an assessment of the student on each of the students’ clinical performance which is comprised of the following 10 items: history taking, physical examination, fund of knowledge, patient management, clinical evaluation and management skills, interpersonal skills, presentation skills, professionalism, cultural/social/systems awareness and initiative and desire to learn. The clerkship director then assigned one of the following grades based on their assessment of the written evaluations: Honors with Distinction (HD), Honors (H), Pass (P), or Fail (F). There was no written examination.

Reviewer comment: 6- It would be better to write the participants section at the beginning of the methods section.

Author response: We agree and have moved up the participants section as previously described.

Reviewer comment: Results:

7- The description of table 2 was not precise and clear.

Author response: We thank the reviewer for this observation and edited the description to state “Paired student grades for the medicine sub-internship, stratified by third year internal medicine clerkship performance.”

Reviewer comment: Discussion and Conclusion:

These sections were written well.

Author response: Thank you for these kind words

---

## [Editor Report · Decision Letter 2]

13 Oct 2022

PONE-D-21-36455R2Pairing medical students on the wards: A multi-site analysis of pairing effect on clerkship performancePLOS ONE

Dear Dr. Miloslavsky,

Thank you for submitting your manuscript to PLOS ONE. After careful consideration, we feel that it has merit but does not fully meet PLOS ONE’s publication criteria as it currently stands. Therefore, we invite you to submit a revised version of the manuscript that addresses the points raised during the review process.

After reviewers point of view, it was decided that authors do minor revision in the light 3rd reviewer's comments

We look forward to receiving your revised manuscript.

Kind regards,

Rano Mal Piryani, MBBS, MCPS, DTCD, MD, Fellowship in Med Education

Academic Editor

PLOS ONE
---

## [Author Response · Author response to Decision Letter 2]

3 Nov 2022

Abstract:

1- Please explain more about the procedure of your work.

Author response: Thank you for this suggestion, we expanded the Methods section of the abstract as follows. “We examined 186 4th year student-pairs during the required medicine sub-internship at 3 hospital sites of Harvard Medical School from 2013-2017. Chi-square tests were used to determine whether pairing impacted the final clerkship grade. Subsequently we examined whether pairing impacted the sub-internship performance stratified by students’ 3rd year core medicine clerkship grade to account for prior performance.”

Reviewer comment: 2- Please write the keywords based on MeSH.

Author response: We have added the keywords as suggested

Reviewer comment: Introduction:

This section was written very well.

Methods:

3- The second sentence was not written clearly; please modify it.

Author response: We agree and have reworded this sentence as follows, “We did not examine other clerkships because students either worked alone or were paired with multiple different partners during the rotation. In addition, other required clerkships utilized shelf exams in their grading, further hindering the evaluation of the pairing effect on performance.”

Reviewer comment: 4- It would be better to write the number of paired students in the setting section.

Author response: Thank you for this suggestion. We merged the setting and participants section with the participants section first to have the number of students examined early in the methods section.

Reviewer comment: 5- Please explain more about the sub-internship grade in the setting section.

Author response: We elaborated on the sub-internship grade as follows. “Sub-internship grades were assigned by the clerkship director at each site based on written evaluations by faculty and residents who worked directly with the student. Each evaluator submitted an assessment of the student on each of the students’ clinical performance which is comprised of the following 10 items: history taking, physical examination, fund of knowledge, patient management, clinical evaluation and management skills, interpersonal skills, presentation skills, professionalism, cultural/social/systems awareness and initiative and desire to learn. The clerkship director then assigned one of the following grades based on their assessment of the written evaluations: Honors with Distinction (HD), Honors (H), Pass (P), or Fail (F). There was no written examination.

Reviewer comment: 6- It would be better to write the participants section at the beginning of the methods section.

Author response: We agree and have moved up the participants section as previously described.

Reviewer comment: Results:

7- The description of table 2 was not precise and clear.

Author response: We thank the reviewer for this observation and edited the description to state “Paired student grades for the medicine sub-internship, stratified by third year internal medicine clerkship performance.”

---

## [Editor Report · Decision Letter 3]

6 Nov 2022

Pairing medical students on the wards: A multi-site analysis of pairing effect on clerkship performance

PONE-D-21-36455R3

Dear Dr. Miloslavsky,

We’re pleased to inform you that your manuscript has been judged scientifically suitable for publication and will be formally accepted for publication once it meets all outstanding technical requirements.

Kind regards,

Rano Mal Piryani, MBBS, MCPS, DTCD, MD, Fellowship in Med Education

Academic Editor

PLOS ONE
---

## [Editor Report · Acceptance letter]

18 Nov 2022

PONE-D-21-36455R3 

Pairing medical students on the wards: A multi-site analysis of pairing effect on clerkship performance 

Dear Dr. Miloslavsky:

I'm pleased to inform you that your manuscript has been deemed suitable for publication in PLOS ONE. Congratulations! Your manuscript is now with our production department. 

Kind regards, 

on behalf of

Dr. Rano Mal Piryani 

Academic Editor

PLOS ONE